# Mapping of Back Muscle Stiffness along Spine during Standing and Lying in Young Adults: A Pilot Study on Spinal Stiffness Quantification with Ultrasound Imaging

**DOI:** 10.3390/s20247317

**Published:** 2020-12-19

**Authors:** Christina Zong-Hao Ma, Long-Jun Ren, Connie Lok-Kan Cheng, Yong-Ping Zheng

**Affiliations:** Department of Biomedical Engineering, The Hong Kong Polytechnic University, Hong Kong, China; czh.ma@polyu.edu.hk (C.Z.-H.M.); longjun.ren@polyu.edu.hk (L.-J.R.); lk-connie.cheng@polyu.edu.hk (C.L.-K.C.)

**Keywords:** back muscle stiffness, spine, elasticity, shear-wave elastography (SWE), tissue ultrasound palpation system (TUPS), reliability, Young’s modulus

## Abstract

Muscle stiffness in the spinal region is essential for maintaining spinal function, and might be related to multiple spinal musculoskeletal disorders. However, information on the distribution of muscle stiffness along the spine in different postures in large subject samples has been lacking, which merits further investigation. This study introduced a new protocol of measuring bilateral back muscle stiffness along the thoracic and lumbar spine (at T3, T7, T11, L1 & L4 levels) with both ultrasound shear-wave elastography (SWE) and tissue ultrasound palpation system (TUPS) in the lying and standing postures of 64 healthy adults. Good inter-/intra-reliability existed in the SWE and TUPS back muscle stiffness measurements (ICC ≥ 0.731, *p* < 0.05). Back muscle stiffness at the L4 level was found to be the largest in the thoracic and lumbar regions (*p* < 0.05). The back muscle stiffness of males was significantly larger than that of females in both lying and standing postures (*p* < 0.03). SWE stiffness was found to be significantly larger in standing posture than lying among subjects (*p* < 0.001). It is reliable to apply SWE and TUPS to measure back muscle stiffness. The reported data on healthy young adults in this study may also serve as normative reference data for future studies on patients with scoliosis, low back pain, etc.

## 1. Introduction

Multiple musculoskeletal disorders can take place in the spine of human beings, including the spinal curvature deformity of adolescent idiopathic scoliosis [1] and chronic low back pain [2]. Both scoliosis [3,4] and low back pain [5] could lead to significant socioeconomic burdens and reduced quality of life in patients. Previous studies have reported that the imbalance of spinal muscles existed in scoliosis patients [6,7] and patients with low back pain [8,9].

The mechanical property of muscle stiffness is an essential factor for maintaining muscle function. It is related to muscle performance during exercise [10] and the joint constraint [11]. Abnormally increased muscle stiffness can be found in a number of musculoskeletal disorders, including spasticity [12], aging [13,14], spinal muscle atrophy [15], low back pain [16], adolescent idiopathic scoliosis [17,18], reduced joint range of motion [19], and reduced lumbar flexion during prolonged sitting [20]. Meanwhile, decreased muscle stiffness may lead to a higher risk of joint dislocation [21] and reduced muscle power [22]. Quantifying muscle stiffness can be helpful for understanding the mechanisms of these musculoskeletal symptoms and pathologies.

Muscle stiffness can be quantified by tissue elasticity (E) and measured by various technologies, including mechanical measurement, ultrasound indentation, and elastography. More recently, the non-invasive and real-time ultrasound shear-wave elastography (SWE) has become a popular and useful tool for assessing muscle stiffness [23,24]. For this technology, the ultrasound probe can induce a focused acoustic force and create a shear wave within the target tissue [25,26]. By capturing the propagation of the shear wave, the speed of the shear wave propagation (c) can be calculated, which can then be squared and multiped by three and the muscle mass density (ρ = 1000 kg/m^3^) to calculate the muscle elasticity (assumed as E = 3ρc^2^) [25]. With this rather new technology, a few previous studies have managed to measure the spinal muscle stiffness using the SWE; however, most of them were limited to the low back region, such as longissimus [27], multifidus [28], and erector spinae [28] of healthy subjects with the sample size being less than 24 participants.

The tissue ultrasound palpation system (TUPS) is another ultrasound-based instrument that can assess muscle stiffness. The probe of TUPS contained both force sensor and ultrasound transducer, which can record the real-time tissue deformation with the conventional B-mode ultrasound image to calculate Young’s modulus or elasticity and acquire the tissue thickness [29]. The TUPS has been applied to evaluate the tissue stiffness of the foot [30] and scar thickness [31]. Two pilot studies have also used TUPS to measure the stiffness of back muscle at L4 level in 12 healthy subjects and 12 patients with low back pain [16], and at L1 and L4 levels in 10 patients with low back pain [29]. The latest updated TUPS system is a handheld and wireless version, which could eliminate the afference of the wires during measurement and make it more feasible to do the measurement in a clinical setting in the future [29].

To date and to the best of the authors’ knowledge, none of the previous studies have used either the SWE or the TUPS to evaluate the distribution of back muscle stiffness, along the thoracic and lumbar spinal regions, in a large number of adults. While both SWE and TUPS have been used to evaluate back muscle stiffness, the sample size has been very small (≤24 participants) and the evaluated region has been limited to the lower back of the spine. The generalization of these findings has been rather limited, and the normative data of muscle stiffness along the spine in different postures have been lacking. While both SWE and TUPS have been used to evaluate back muscle stiffness, the comparison of these two instruments on measuring back muscle stiffness was still scarce.

To address the above-mentioned issues, the current study aimed to (1) measure the back muscle stiffness along the spine at the levels of T3, T7, T11, L1, and L4 with both SWE and TUPS in both lying and standing postures in 64 healthy adults (*mapping the back muscle stiffness* with *large sample* size, the effect of *levels*); (2) compare the measured results of muscle stiffness between SWE and TUPS in terms of *reliability* and *relationship*; (3) identify the difference of distribution in muscle stiffness along the spine between the standing and lying postures (effect of *posture*), and (4) determine if the gender factor influenced the distribution of muscle stiffness along the spine in both standing and lying postures (effect of *gender*). The levels of T3, T7, T11, L1, and L4 were selected to reflect the muscle stiffness along the upper, middle, and lower thoracic and the upper and lower lumbar spinal regions [32].

## 2. Materials and Methods

### 2.1. Subjects

In total, sixty-four healthy young adults (32 males and 32 females) aged between 18 and 30 years were recruited. Subjects were excluded if they had low back pain within the last three months before the study; scoliosis; muscular disease of limbs or spine; and/or history of bone disease, fracture, surgery, or malformation at the spinal region. A Registered Physiotherapist verified the inclusion and exclusion criteria via subject interview and physical examination before the data collection.

Subjects were instructed not to have any vigorous exercise two days before the experiment, to avoid possible muscle fatigue and altered muscle stiffness that may affect the experimental results [33]. They should also avoid muscle relaxants and alcohol before the experiment, along with some other drugs. During the experiment, if subjects experienced any discomfort, the experiment would be stopped immediately with the condition been recorded. Ethical approval was granted by the authority of the local university (HSEARS20180122004). Written informed consent was signed and obtained from all subjects before the experiment.

### 2.2. Instruments for Measuring Back Muscle Stiffness

#### 2.2.1. Wireless Hand-Held Tissue Ultrasound Palpation System (TUPS)

A newly-updated hand-held tissue ultrasound palpation system (TUPS), with a probe (7.5 MHz 128-elements ultrasound transducer with a 20 N in-series load cell) wirelessly connected to a laptop via Wi-Fi, was used to evaluate the back muscle stiffness (referred to as “TUPS stiffness” in this paper) and the thickness of soft tissues [29]. The probe sampled the ultrasound image and force data simultaneously and transmitted them to a laptop in real-time [29,34,35,36]. The frequency of the system was 12 Hz. For each measurement, five compression-release cycles within a duration of 10 s were performed to collect data. The thickness of soft tissue along the spine was also measured by the TUPS system.

#### 2.2.2. Ultrasonic Scanner with Shear-Wave Elastography (SWE)

A commercially available multi-wave ultrasonic scanner (version 10.0; Super-Sonic Imagine, Aix-en-Provence, France) coupled with a convex probe (SuperCurved 6-1, Super-Sonic Imagine, Aix-en-Provence, France) in shear-wave elastography (SWE) and musculoskeletal (MSK) mode was used to evaluate the back muscle stiffness. For each measurement, a 10-s video with approximately 10 frames of ultrasound images was recorded and exported in “MP4” format, after the color map of stiffness was maintained as homogeneously as possible.

A developed Matlab script (Version 2016b, MathWorks, MA, USA) was used to process the stiffness data of the exported video. Firstly, the region of interest (ROI) was selected as the largest muscle area that avoided bone, fascia, or subcutaneous tissue. Secondly, the artifact pixel (showing no color in the color map or saturate at 300 kPa) within the ROI was excluded for data analysis. Thirdly, the Matlab image processing script converted each available pixel of the color map into a value of stiffness, based on the color scale. Finally, the mean value of the stiffness from the captured 10 frames was calculated to obtain the stiffness of each measurement (referred to as “SWE stiffness” in this paper) for further statistical analysis.

Pilot studies were conducted to evaluate the validity of the introduced data analyzing method with the developed Matlab image processing script. It revealed that the results generated by this Matlab script were similar to those from the Aixplorer scanner software (Q-BoxTM) as used in [27,28,37].

### 2.3. Experimental Procedure

Before the measurement, the spinal processes at the T3, T7, T11, L1, and L4 levels were located by palpation and ultrasound B-mode image and then marked with a water-insoluble eyeliner. The bilateral muscle belly that parallels with the spinal process was located for measuring the back muscle stiffness as suggested in [38].

During the experiment, all measurements were conducted in lying posture first, followed by the standing posture for both the instruments of TUPS and SWE. For the *lying* posture, subjects were prone with their faces in the hole of a massage bed and their upper limbs along the trunk. For the *standing* posture, subjects stood in front of a supporting frame to eliminate the possible influence generated by the compression-release of the TUPS’ probe during the measurement on posture. Subjects were instructed not to resist the compression force generated by the TUPS voluntarily, but to reply on the supporting frame instead (Figure 1). Subjects were instructed to breathe naturally, put their weight equally on both feet, and maintain their heads in a neutral position during the measurement. A male assessor and two female assessors conducted the measurements for all male and female subjects, respectively. Assessors would instruct the subjects to adjust posture if subjects stood asymmetrically.

### 2.4. Reliability Test

Since three assessors (1 male and 2 females) conducted the measurements in this study, a reliability test was conducted to determine the reliability of the measurements prior to the start of the main experiment. The measurement and reliability tests were conducted on fourteen male subjects. During the test, the muscle stiffness of the left and right sides at the T7 and L1 levels were measured three times by each assessor in the lying position.

### 2.5. Data and Statistical Analysis

Statistical analysis was conducted using the SPSS (Version 24, SPSS Inc, Chicago, IL, USA). The muscle stiffness value of the left and right sides was averaged for statistical analysis. The percentage change of muscle stiffness from lying to standing posture was also calculated. Intra-rater and inter-rater reliability of the muscle stiffness measurements were examined with the Intraclass Correlation Coefficient (ICC (3,1)) with a 95% confidence interval (95% CI). Three-way mixed ANOVA with post-hoc pairwise comparison was conducted to determine the main effects of (1) *posture* (lying vs. standing), (2) *level* (T3, T7, T11, L1 vs. L4), and (3) *gender* (male vs. female), as well as the interaction effect on muscle stiffness. The Pearson correlation test was performed to examine the relationship between the two measurement techniques of SWE and TUPS. The significance level was set at 0.05. The effect size (η_p_^2^) for each parameter was also presented.

## 3. Results

A total of 64 subjects (32 males and 32 females, aged 23.5 ± 2.9 years, height 166.4 ± 8.5 cm, weight 60.1 ± 10.3 kg, and BMI 21.6 ± 2.5) participated in this study. Initially, seventy-five subjects were screened for this study. Among them, two subjects were excluded due to scoliosis, and four subjects were excluded due to uncomfortableness during the experiment. The data of five subjects were discarded due to the technological issue of hard disk failure where the data cannot be retrieved.

### 3.1. Intra-/Inter-Rater Reliability of Measurements

Good intra-rater reliability [*TUPS*: ICC = 0.822 (95% CI 0.632 to 0.962) at T7, and ICC = 0.905 (95% CI 0.704 to 0.969) at L1; *SWE*: ICC = 0.881 (95% CI 0.631 to 0.962) at T7, and ICC = 0.879 (95% CI 0.625 to 0.961) at L1] and good inter-rater reliability [*TUPS*: ICC = 0.742 (95% CI 0.368 to 0.910) at T7, and ICC = 0.836 (95% CI 0.602 to 0.943) at L1; *SWE*: ICC = 0.731 (95% CI 0.340 to 0.906) at T7, and ICC = 0.781 (95% CI 0.462 to 0.924) at L1] for both measurement instruments were identified in this study (*p* < 0.05). Higher intra-rater reliability of TUPS than SWE except at T7 level, and higher inter-rater reliability of TUPS than SWE were also found.

### 3.2. Muscle Stiffness Measured by TUPS

The measured muscle stiffness by TUPS is summarized in Table 1 and Figure 2. Significant main effects of gender (*p* < 0.001, η_p_^2^ = 0.292) and level (*p* < 0.001, η_p_^2^ = 0.601), and significant interaction effect among three factors (*p* < 0.001, η_p_^2^ = 0.069) were found.

#### 3.2.1. Effect of Gender

The results of post-hoc comparison revealed that the TUPS stiffness was significantly larger in male subjects than that of female subjects for all five levels (*p* < 0.001, η_p_^2^ = 0.292).

#### 3.2.2. Effect of Level

The TUPS stiffness was found to be significantly different among the five different levels (T3: 163.4 ± 25.8 kPa, T7: 142.5 ± 20.9 kPa, T11: 138.1 ± 18.4 kPa, L1: 165.4 ± 25.6 kPa, and L4: 217.9 ± 33 kPa), except the two pairwise comparisons of T3 vs. L1 and T7 vs. T11. More specifically, the muscle stiffness significantly decreased from T3 to T7 level (*p* < 0.05), significantly increased from T11 to L1 level (*p* < 0.05), and significantly increased from L1 to L4 level (*p* < 0.05). The muscle stiffness at L4 was found to be significantly largest (*p* < 0.05); meanwhile, the muscle stiffness at T11 tended to be the smallest, but did not reach a significant level.

#### 3.2.3. Effect of Posture

No significant difference in TUPS stiffness between the two postures was found (lying: 163.4 ± 25.8 kPa and standing: 163.4 ± 25.8 kPa, *p* = 0.707, η_p_^2^ = 0.002). Meanwhile, upon looking into the TUPS stiffness at each level, significantly larger TUPS stiffness at T3, L1, and L4 during standing posture than lying in male subjects existed, and a reversed trend of significantly larger TUPS stiffness at T3, T7, T11, and L1 during lying posture than standing in female subjects existed.

As shown in Figure 3, the percentage difference of changes in TUPS stiffness was significantly larger in male subjects than female subjects at all four levels (*p* ≤ 0.005). More specifically, the percentage change at the L4 level appeared to be the largest (48.5% for males and 4.9% for females), and that of T7 was found to be the smallest (−2.9% for males and −30.4% for females).

### 3.3. Muscle Stiffness Measured by SWE

The measured muscle stiffness by SWE is summarized in Table 2 and Figure 4. Significant main effects of gender (*p* = 0.030, η_p_^2^ = 0.074) and posture (*p* < 0.001, η_p_^2^ = 0.772), and significant interaction effect only between gender and posture (gender*posture: *p* = 0.016, η_p_^2^ = 0.090) were found.

#### 3.3.1. Effect of Gender

The results of post-hoc comparison revealed that the SWE stiffness was also significantly larger in male subjects than that of female subjects for all five levels (*p* = 0.003, η_p_^2^ = 0.074).

#### 3.3.2. Effect of Posture

The SWE stiffness in lying posture was found to be significantly smaller than that of standing posture for both genders and for all five different levels (lying: 22.5 ± 1.6 kPa, and standing: 41.3 ± 2.2 kPa, *p* < 0.001, η_p_^2^ = 0.772).

As shown in Figure 5, the percentage difference of changes in SWE stiffness fluctuated, and no significant difference among different levels was found. While a significantly larger change in SWE stiffness in males than females at the L1 level was found (*p* = 0.013), no other significant difference between the two genders was found.

#### 3.3.3. Effect of Level

No significant difference in SWE stiffness among the five different levels was found (T3: 29.7 ± 12.5 kPa; T7: 32.9 ± 11.6 kPa; T11: 33.5 ± 12.2 kPa; L1: 32.2 ± 11.4 kPa and L4: 31.1 ± 9.0 kPa).

### 3.4. Change of Soft Tissue Thickness from Lying to Standing Posture

As shown in Figure 6, a significantly moderate correlation between soft tissue thickness and posture was found at all five levels (r = 0.304, *p* < 0.001). Significant main effects of posture (*p* = 0.001), level (*p* < 0.001), and gender (*p* < 0.001), and no significant interaction effect among these factors were found. The soft tissue thickness was also found to be significantly larger in standing posture than lying posture at five levels (*p* = 0.001). Similar to the distribution of muscle stiffness at different levels, the soft tissue thickness was also found significantly decreased from T3 to T7 level, significantly increased from T11 to L1 level, and significantly increased from L1 to L4 level (*p* < 0.001). The soft tissue thickness at the L4 level was found to be significantly largest (*p* < 0.001), and the thickness at the T7 level was found to be the smallest (*p* < 0.001).

### 3.5. Relationship between the SWE and TUPS Measurement Techniques

Table 3 summarizes the results of Pearson’s correlation coefficient (r) between the SWE and TUPS measurement techniques in lying and standing positions. Significantly moderate correlations between the SWE and TUPS measurement techniques were observed at T3 in the lying posture (r = −0.294, *p* = 0.018), L1 in the standing posture (r = 0.390, *p* = 0.001), and L4 in the standing posture (r = 0.358, *p* = 0.004). Low correlations between the two measurement techniques were observed at the remaining levels and postures, but the correlations were not significant.

## 4. Discussions

To our knowledge, this is the very first study investigating the distribution of back muscle stiffness along the spine, with the stiffness measurements from both the SWE and TUPS, in healthy young adults. Several significant effects of posture, level, and gender on back muscle stiffness were identified in this study. The relationship and reliability of the SWE and TUPS measurement techniques were also investigated and established.

### 4.1. Intra-/Inter-Rater Reliability of Measurements

Good intra-rater and good inter-rater reliability for both measurement instruments of TUPS and SWE were observed in this study. This is in accordance with previous studies on the reliability of the TUPS [30,31,34] and SWE [39]. Additionally, the handheld and wireless setting of the TUPS also makes it more convenient to be applied in clinical settings [29] and not be restrained by the experiment [40] in the future.

### 4.2. Effect of Level (T3, T7, T11, L1 vs. L4)

This study has uncovered that regardless of the lying or standing posture, the back muscle at the L4 level has the largest TUPS stiffness value among the thoracic and lumbar regions that covered the commonly affected sites of scoliosis and low back pain. This helps explain why chronic low back pain has been commonly found in the L3-L4 region in previous studies [2,29]. The observed large stiffness values at L4 level in healthy adults in this study might help explain the cause/development of the low back pain pathology later on in middle-aged or older adults. It is likely that the greater muscle stiffness at the L4 level might be one potential cause of scoliosis or low back pain, however, future studies shall be conducted to explore and clarify this. Upon measuring the back muscle stiffness of both thoracic and lumbar regions in large samples (*n* = 64), the results of this study could also act as a normative database for future studies on scoliosis, low back pain, and other spinal musculoskeletal disorders.

The TUPS stiffness was found to decrease from T3 to T7, and then increase from T11 to L4 level. This is in line with previous studies suggesting that the muscle stiffness at L4 was larger than L1 in patients with low back pain [29], patients undergoing spinal surgery [41], and healthy young adults [27]. This could be explained by the curvature of the spine since the thoracic spine is in kyphosis position, which might stretch the back muscles; and the lumbar spine is in the lordosis region, which might compress the back muscles. Future studies, preferably using the real-time ultrasound images with large resolutions that can observe muscle fibers, are needed to further explore and look into the underlying mechanism of the observed different changes of back muscle stiffness at the thoracic and lumbar regions. It is also worthwhile to investigate if the observed change in muscle stiffness is a physiological phenomenon or the cause of pathology in future studies.

Unlike TUPS, no statistically significant difference in levels in SWE was observed in this study. This might be explained by the different underlying mechanisms of evaluating muscle stiffness by TUPS and SWE. While TUPS could not differentiate the multiple layers of soft tissues, SWE could specifically locate a certain muscle area by choosing the region of interest (ROI) on an ultrasound image. Most previous studies have compared the difference between lying and upright positions and/or between resting and contraction conditions of a single muscle group [42,43]. Unfortunately, few previous studies have used SWE to map the muscle stiffness along the spine, which makes it rather difficult to compare the current findings in this study with previous studies. The lack of previous findings might be because SWE is a rather new technology, and more studies shall be conducted in the future to enable the synthesis of information on spinal muscle stiffness as evaluated by SWE.

### 4.3. Effect of Gender (Female vs. Male)

Back muscle stiffness of male subjects was found to be larger than that of female subjects in both lying and standing postures as evaluated by TUPS and SWE. This accords with previous studies on muscle stiffness of knee extensor in healthy young athletes [44] and knee flexors in healthy young adults [45]. Male and female adults have different anatomical structures generally, such as the distribution and mass of body fat and muscles at the back of the trunk [46]. This might help explain the observed difference in back muscle stiffness between the two genders in this study. While most of the previous studies focused on limb muscle stiffness [44,45], this study provides more information and evidence about the effect of gender on back muscle stiffness along the thoracic and lumbar spine.

### 4.4. Effect of Posture (Lying vs. Standing)

SWE stiffness was found to be larger in standing posture than lying posture, while significantly larger TUPS stiffness at T3, L1, and L4 during standing in male subjects and that at T3, T7, T11, and L1 during lying in female subjects were found. Meanwhile, significant percentage changes in muscle stiffness from the lying to the standing posture was found for TUPS measurement, but not for the SWE measurement in this study, even the pattern/trend of changes appeared to be the same. The finding of significantly larger TUPS stiffness at the L4 level during standing in male subjects was in line with a previous study on male patients with low back pain [16]. The certain difference regarding the effect of posture between the SWE and TUPS measurements could be due to the different measuring mechanisms of these two instruments. The SWE assumed the muscle mass density (ρ = 1000 kg/m^3^) to be a constant value when calculating the stiffness [25], and TUPS used the real-time tissue deformation with a conventional B-mode image to calculate the stiffness [29]. The soft tissue thickness has changed from lying to standing position, which might affect the muscle mass density and thus affect the SWE stiffness measurement. The SWE stiffness has also been suggested to be minimally associated with TUPS stiffness [47]. Additionally, the soft tissue thickness was found to be correlated with the different postures of lying and standing, and the changing pattern of soft tissue thickness also appeared to be similar to that of muscle stiffness at different levels as measured by TUPS in this study. 

The different changes in TUPS stiffness from lying to standing postures between male and female subjects might be due to the different anatomical structures between the two genders. The TUPS used the tissue deformation during the compression-release cycles to acquire the stiffness [29]. It is reasonable to expect that the mass of body fat and breast may affect more of the back muscle elongation and contraction, especially in the upper trunk in female subjects than that of male subjects. Previous studies have also reported the increased anterior-posterior shear forces in females and decreased forces in males in response to stress [48], and the different trunk muscle geometry between the two genders [49]. All these findings may help explain the measured different changes from lying to standing posture between the SWE and TUPS measurements and between the two genders. Further studies, preferably in vitro studies with better control of the experimental setting for more robust results, are still needed to look into this issue and identify the exact cause.

### 4.5. Relationship between the SWE and TUPS Measurement Techniques

This study observed significantly moderate correlations between the SWE and TUPS measurement techniques at the T3 level in the lying position, L1 level in the standing position, and L4 level in the standing position only. This might be caused by the different underlying mechanisms to measure muscle stiffness by the two techniques. Additionally, while TUPS could not differentiate the multiple layers of soft tissues, SWE could specifically locate a certain muscle area by choosing the region of interest (ROI) on an ultrasound image. The muscle contraction and the overlying tissue might influence the SWE and TUPS stiffness measurements. The observed correlation may also contribute to the presented results and explain the difference in muscle stiffness as measured by the two techniques at the same location. For example, the TUPS stiffness values ranged between 104 kPa and 303 kPa, while SWE stiffness values ranged between 17 kPa and 37 kPa. This implies that when applying these two techniques to evaluate the muscle stiffness, the relative changes in measured values might be more meaningful for muscle assessment than that of absolute values.

### 4.6. Implications and Outlook

The results of this study will not only enlarge our understanding of back-muscle stiffness, but also provide us with insights about the possible cause and mechanism of scoliosis and low back pain. The findings of this study inspire future efforts to investigate if the observed change in muscle stiffness is a physiological phenomenon or the cause of pathologies of scoliosis and low back pain. The results may also serve as normative data about back muscle stiffness in healthy young adults, which could be useful in lots of healthcare areas in the future. Attention should be paid to the fact that the normative values might be different for different age groups, and only healthy young people aged between 18–30 years were evaluated in this study. The obtained values on muscle stiffness could be treated as a reference, but it might not be appropriate to directly generalize such values to patients with scoliosis or lower back pain. This study can be the first step for further studies investigating the distribution of back muscle stiffness along the spine in different populations, age groups, and patient groups. As the introduced experimental protocol has been evaluated to be reliable, this study also enables future studies to apply similar protocols on patients with scoliosis, low back pain, and other spinal musculoskeletal disorders.

### 4.7. Best Practice Recommendations for the SWE and TUPS Measurement Techniques

Good knowledge of the mechanism of the SWE and TUPS measurement techniques and careful consideration of the available experimental/clinical settings are needed to enable the best practice for the two techniques. It is recommended to apply the SWE to measure the muscle stiffness when differentiating and evaluating different layers of muscles and different regions of the same muscle, since the region of interest (ROI) of SWE can be easily customized to various shapes and sizes at various locations during the measurement. With the advantages of being portable, with a wireless connection, and occupying a small space, the TUPS would be more helpful to generally evaluate the muscle stiffness at certain anatomical locations, especially for screening purposes in labs/clinics with limited space or even in an outdoor environment [29,40]. The TUPS can be used to do the screenings first, and when abnormal muscle stiffness is identified, the SWE can then be applied to examine the exact cause by specifically locating various regions.

### 4.8. Limitations

While mapping the muscle stiffness at the thoracic and lumbar regions could already provide plenty of information, the data on cervical and sacral regions were not involved in this study. Future studies could consider expanding this mapping strategy to provide a more comprehensive picture of spinal muscle stiffness. During the experiment, subjects were instructed to stand symmetrically and put equal weight on both feet, supplemented by the observation of accessors. Future studies could consider putting pressure sensors and/or a pressure mat under the subject’s feet to control the symmetric weight distribution more objectively. While subjects were instructed to breathe naturally during the experiment, it should be noted that the influence of breath dynamics on the measured results remained unclear. Future studies are needed to understand this issue. The muscle activity was unfortunately not measured as a covariable to explain muscle stiffness changes, which can also be considered in future investigations.

It shall be noted that the SWE and TUPS measurements might be sensitive to other factors besides the mechanical properties. A recent paper reported that indirect tissue stiffness measurements are often sensitive to the mechanical properties, geometrical dimensions, and tensional state of the tissue [50]. From a mechanical perspective, tissue elasticity is defined as a ratio of strain (strain = F/A) and elongation (dl/L), and can be measured directly using mechanical measurements. Meanwhile, ultrasound and elastography do not measure strain and elongation directly. These indirect measurements are indirect estimations of the stiffness. When standing, the muscles might contract and thereby influence the stiffness measurements. For the research community, it is essential to have a standard definition and understanding of muscle stiffness. Further studies are still needed to investigate, understand, and clarify this issue.

## 5. Conclusions

This study mapped the back muscle stiffness along the thoracic and lumbar spine, with both SWE and TUPS, in both lying and standing postures in healthy male and female adults. It identified that both SWE and TUPS are reliable in measuring back muscle stiffness. Significant effects of level, gender, and posture on back muscle stiffness were identified. This may provide some insights regarding the underlying mechanism of why the common sites of low back pain take place at a certain spinal region and facilitates future research on other spinal musculoskeletal disorders.

## Figures and Tables

**Figure 1 sensors-20-07317-f001:**
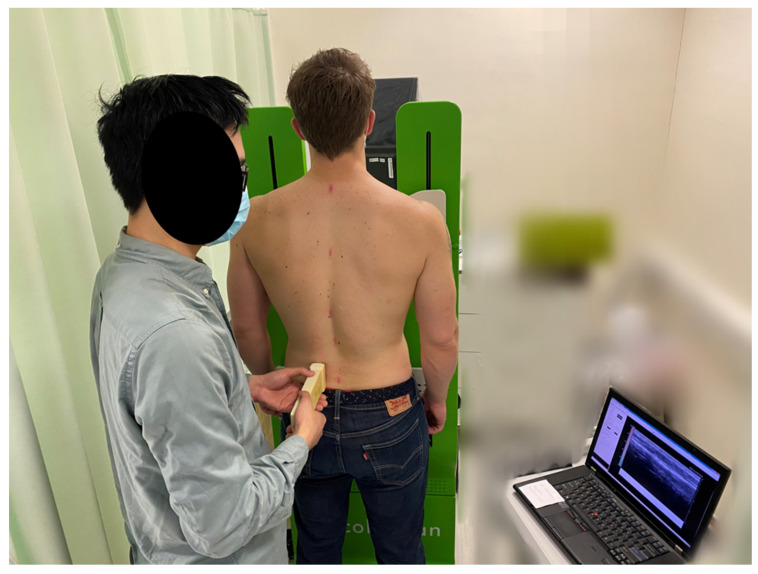
Illustration of the assessment in standing position with a supporting frame.

**Figure 2 sensors-20-07317-f002:**
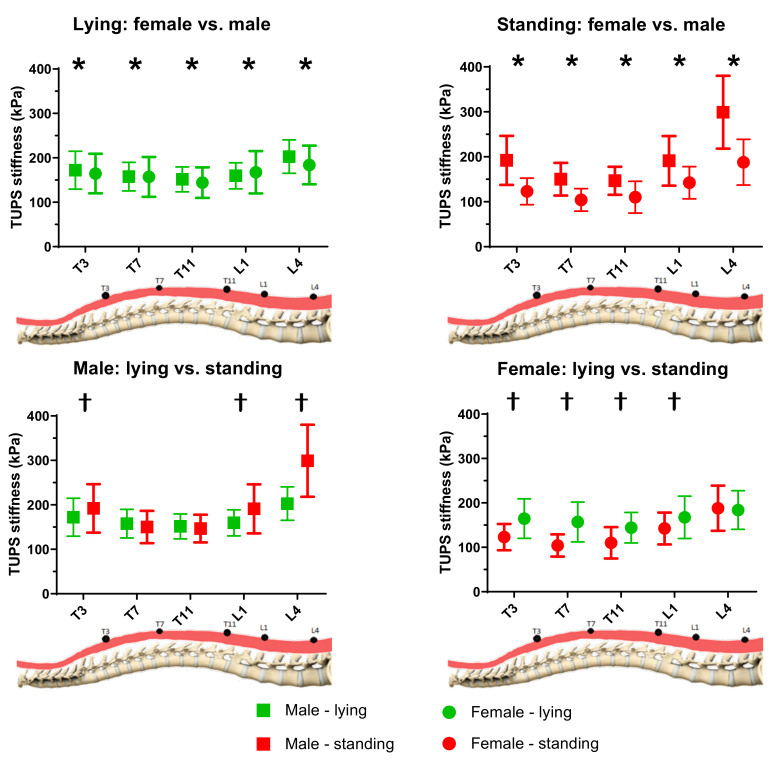
TUPS stiffness at different levels of two genders in lying and standing postures (*n* = 64). * Significantly difference in gender; † Significant difference in posture.

**Figure 3 sensors-20-07317-f003:**
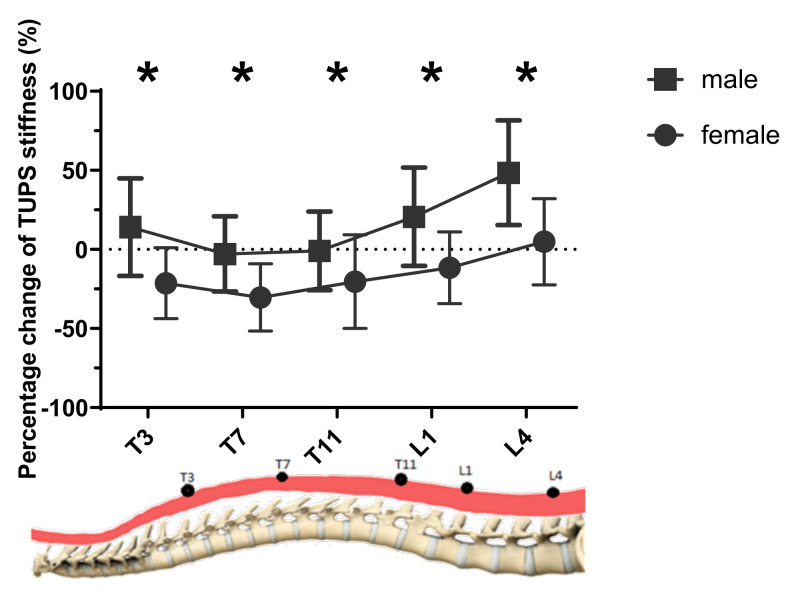
The percentage change of TUPS stiffness from lying to standing posture at different levels of two genders (*n* = 64). * Significant differences existed in gender.

**Figure 4 sensors-20-07317-f004:**
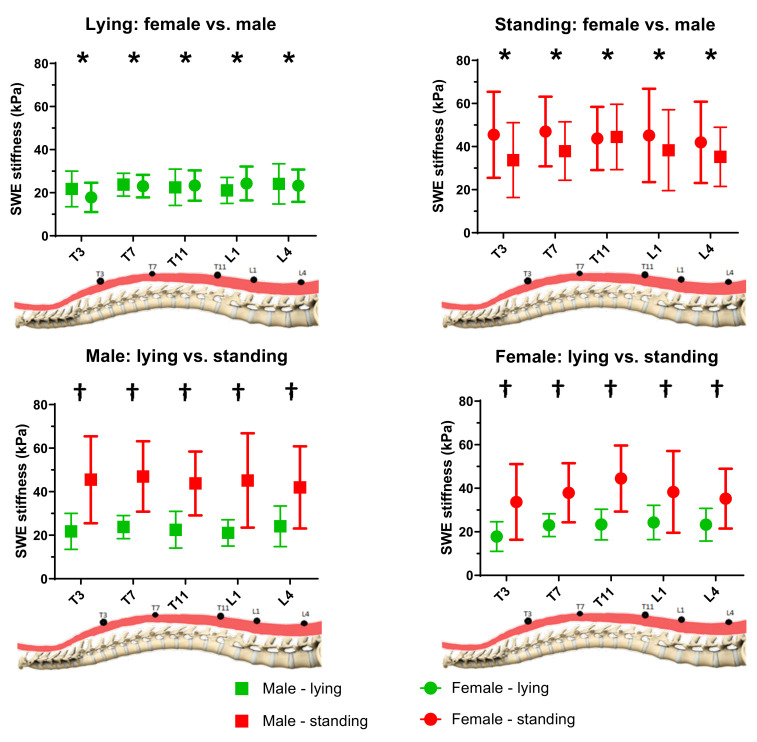
SWE stiffness at different levels of two genders in lying and standing postures (*n* = 64). * Significant differences existed in gender; † Significant differences existed in posture.

**Figure 5 sensors-20-07317-f005:**
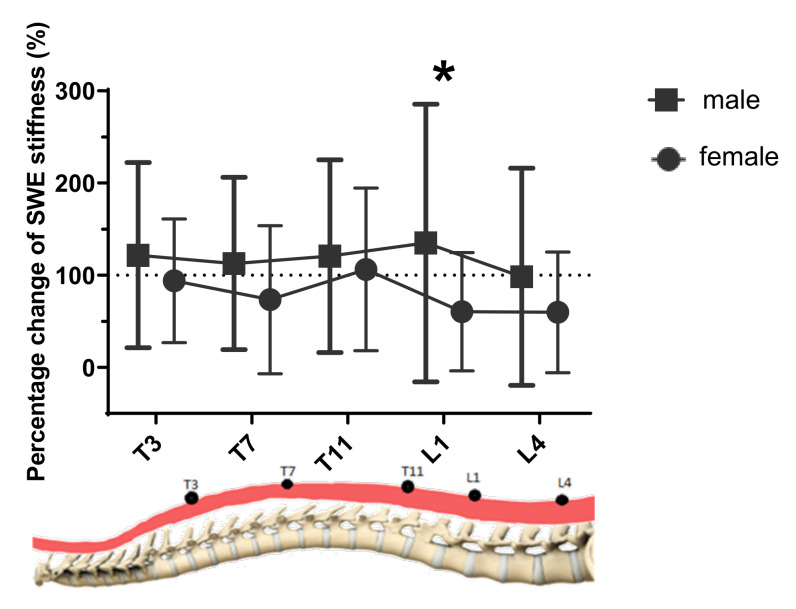
The percentage change of SWE stiffness from lying to standing posture at different levels of two genders (*n* = 64). * Significant differences existed in gender.

**Figure 6 sensors-20-07317-f006:**
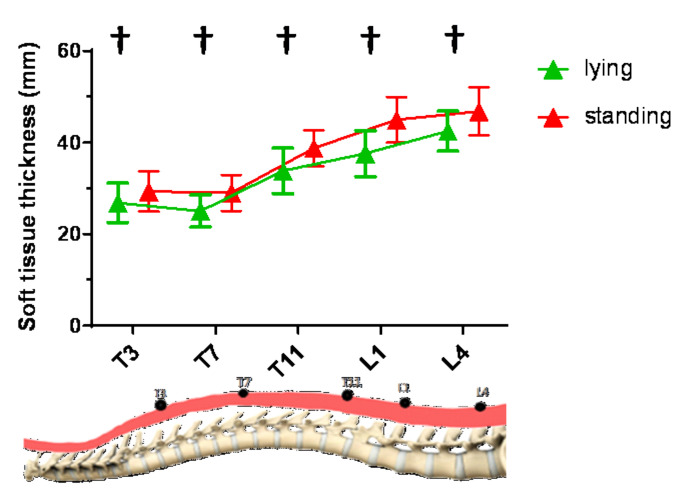
The soft tissue thickness at different levels in lying and standing postures among subjects (*n* = 64). † Significant differences existed in posture.

**Table 1 sensors-20-07317-t001:** The tissue ultrasound palpation system (TUPS) stiffness in kPa (mean ± SD) (*n* = 64).

	Lying		Standing			
Level	Male (*n* = 32)	Female (*n* = 32)	Male (*n* = 32)	Female (*n* = 32)	*p*-Value	Partial Eta Squared, η_p_^2^
**T3**	172.0 ± 42.8 †	166.5 ± 46.4 †	192.3 ± 55.7 *†	122.7 ± 27.8 *†	Main effect:	
					Gender: <0.001	0.292
**T7**	157.8 ± 32.5	155.8 ± 45.2 †	150.8 ± 36.9 *	104.9 ± 26.7 *†	Posture: 0.707	0.002
					Level: <0.001	0.601
**T11**	153.3 ± 27.0	144.4 ± 36.1	148.3 ± 32.0 *	111.3 ± 37.6 *	Interaction effect:	
					Gender * level:<0.001	0.111
**L1**	162.1 ± 29.0 †	168.3 ± 48.4 †	194.7 ± 55.7 *†	137.5 ± 36.0 *†	Gender * posture: <0.001	0.481
					Posture * level: <0.001	0.355
**L4**	205.5 ± 36.2 †	182.7 ± 45.2 †	303.3 ± 80.6 *†	180.5 ± 46.7 *†	Gender * level * posture: <0.001	0.069

* Indicates significant gender difference at a certain level and posture. † Indicates significant posture difference at a certain level for each gender.

**Table 2 sensors-20-07317-t002:** The shear-wave elastography (SWE) stiffness in kPa (mean ± SD) (*n* = 64).

	Lying		Standing			
Level	Male (*n* = 32)	Female (*n* = 32)	Male (*n* = 32)	Female (*n* = 32)	*p*-Value	Partial Eta Squared, η_p_^2^
**T3**	21.8 ± 8.3 *†	17.8 ± 6.8 *†	45.5 ± 20.0 *†	33.7 ± 17.4 *†	Main effect:	
					Gender: 0.030	0.074
**T7**	23.8 ± 5.3 †	23.1 ± 5.2 †	47.0 ± 16.2 *†	37.9 ± 13.6 *†	Posture: <0.001	0.772
					Level: 0.120	0.029
**T11**	22.5 ± 8.4 †	23.3 ± 7.1 †	43.7 ± 14.7 †	44.5 ± 15.1 †	Interaction effect:	
					Gender * level: 0.080	0.033
**L1**	21.1 ± 6.0 †	24.3 ± 7.9 †	45.2 ± 21.6 †	38.3 ± 18.8 †	Gender * posture: 0.016	0.090
					Posture * level: 0.171	0.025
**L4**	24.1 ± 9.3 †	23.3 ± 7.5 †	42.0 ± 18.9 †	35.2 ± 13.7 †	Gender * level * posture: 0.360	0.017

* Indicates a significant gender difference at certain levels and posture. † Indicates a significant posture difference at certain levels of each gender.

**Table 3 sensors-20-07317-t003:** Relationship between the SWE and TUPS measurement techniques (*n* = 64).

	Pearson’s Correlation Coefficient (r)	
Level	Lying	Standing
T3	−0.294 *	0.111
T7	0.196	0.232
T11	−0.141	0.166
L1	0.020	0.390 *
L4	−0.233	0.358 *

* Significantly correlation existed.

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
