# Peer review of "Mapping of Back Muscle Stiffness along Spine during Standing and Lying in Young Adults: A Pilot Study on Spinal Stiffness Quantification with Ultrasound Imaging"

_sensors, 2020, doi:10.3390/s20247317_

Round 1
Reviewer 1 Report
Thank you for opportunity to review the paper entitled “Mapping of back muscle stiffness along spine during standing and lying in young adults: A pilot study on spinal stiffness quantification with ultrasound imaging”. The aim of the study was to introduced a new protocol of measuring bilateral back muscle stiffness along the thoracic and lumbar spine with ultrasound shear-wave elastography (SWE) and tissue ultrasound palpation system (TUPS) in lying and standing postures. The study was performed on 64 healthy adults. Based on the results obtained from the conducted studies, the authors conclude that both measurement (SWE and TUPS) have good reliability. Back muscle stiffness at L4 is largest, is larger in men than in women, and larger in standing position then lying position. The authors also point out that these data may be normative values in the future for patients with scoliosis or back pain, etc.
The article is interesting and addresses the important issue of normative values in the study of muscle stiffness. It also shows how muscle stiffness is distributed in the thoracic and lumbar spine depending on the position and gender. However, in my opinion, some conclusions go too far. Doubts are raised above all by the statement that the obtained values can be used as normative in low back pain or scoliosis. The authors themselves write in line 44 that muscle stiffness changes with age. Young people between 18-30 years of age were examined, but scoliosis affects children and adolescents, and low back pain especially the elderly. Could these results for these groups be the norm?
Introduction
In my opinion, the first paragraph of the introduction is not related to the topic of the work.
The second paragraph could be enriched with works on the abdominal muscles, which is also related to the lumbar region e.g.:
https://pubmed.ncbi.nlm.nih.gov/32868027/
Line 79: What is the rationale for selecting T3, T7, T11, L1 and L4 levels?
Materials and Methods
Line 89-90: Who and how verified the exclusion criteria?
Line 130: If the muscles on both sides were examined, why is there no information anywhere whether the values of CUPS and SWE in the comparison of the sides were not significantly different?
Line 135: Pictures during the examination would be helpful, especially while standing (supporting frame). Was the supporting frame also during the SWE study?
Line 137: How was symmetric weight distribution controlled? Just a word command? How can you be sure that the people were standing symmetrically?
Results
Line 238: How to explain changes in muscle stiffness in TUPS and their absence in SWE when tested at different levels? The authors explain this by a change in the curvature of the spine. Was there no change in curvature in the SWE study?
Discussion
Line 265: The authors use the words stiffness and elasticity as synonyms. This is misleading and may be misunderstood.
Line 266: Is it possible to explain that it is the cause of scoliosis or back pain only because there is greater muscle stiffness at the L4 level?
Line 274-277: If the authors explain the change in muscle stiffness with changes in the curvature of the spine, does it mean that it is a physiological phenomenon? Could this be the cause of pathology?
Line 288: Elasticity was larger or muscle stiffness was larger?
Line 314: Do these studies really explain the cause and mechanism of scoliosis and back pain?
Line 315: I have already written about the normative values. I think that these values will be different for different age groups.
Line 324: There are no other limitations of this research?
Conclusions
Line 332: The last sentence in the conclusions is an abuse. It does not follow from the results achieved.
Author Response
Thank you very much for the efforts and expertise invaluable comments to improve this article. This article has been revised accordingly. Below please kindly find the point-to-point response and detailed description of revisions in the manuscript.
We agree with the comments about generalizing the current findings of this study, which is on healthy young adults, to patients with scoliosis and low back pain. The relevant description has been revised and more clarification has been added to avoid misunderstanding and misguidance to future readers as follows:
- on page 13, line 295-297, stating: “…It is likely that the greater muscle stiffness at the L4 level might be one potential cause of scoliosis or low back pain, however, future studies shall be conducted to explore and clarify this…”
- on page 14-15, line 379-385, stating: “…Attention shall be paid to the fact that the normative values might be different for different age groups, and only healthy young people aged between 18-30 were evaluated in this study. The obtained values on muscle stiffness could be treated as a reference, but it might not be appropriate to directly generalize such values to patients with scoliosis or lower back pain. This study can be the first step for further studies investigating the distribution of back muscle stiffness along the spine in different populations, age groups, and patient groups…”
Introduction
In my opinion, the first paragraph of the introduction is not related to the topic of the work.
The first paragraph mainly describes some background information of spine. It has been revised and shortened in a way that summarize the common spinal musculoskeletal disorders briefly on page 1, line 31-39 as follows: “Multiple musculoskeletal disorders can take place at spine in human beings, including the spinal curvature deformity of adolescent idiopathic scoliosis [1] and chronic low back pain [2]. Both scoliosis [3, 4] and low back pain [5] could lead to significant socioeconomic burdens and reduced quality of life in patients. Previous studies have reported that the imbalance of spinal muscles existed in scoliosis patients [6, 7] and patients with low back pain [8, 9].”
The second paragraph could be enriched with works on the abdominal muscles, which is also related to the lumbar region e.g.: https://pubmed.ncbi.nlm.nih.gov/32868027/
As suggested, the description has been enriched with works on the abdominal muscles cited on page 2, line 44-46, stating: “Abnormally increased muscle stiffness can be found in a number of musculoskeletal disorders, including … adolescent idiopathic scoliosis [21, 22] …”
Line 79: What is the rationale for selecting T3, T7, T11, L1 and L4 levels?
The levels of T3, T7, T11, L1 and L4 were selected to reflect the muscle stiffness along the upper, middle, and lower thoracic and the upper and lower lumbar spinal regions. To clarify and support this protocol, a previously published literature (Buell et al, 2019) has been added and cited to the manuscript. Such clarification has been added on page 2, line 86-87, stating: “…The levels of T3, T7, T11, L1 and L4 were selected to reflect the muscle stiffness along the upper, middle, and lower thoracic and the upper and lower lumbar spinal regions [22] …”
Materials and Methods
Line 89-90: Who and how verified the exclusion criteria?
The exclusion criteria were verified by a registered physiotherapist by subject interview and physical examination. Such statements have been added on page 3, line 94-95, stating: “…A Registered Physiotherapist has verified the inclusion and exclusion criteria via subject interview and physical examination before the data collection…”
Line 130: If the muscles on both sides were examined, why is there no information anywhere whether the values of CUPS and SWE in the comparison of the sides were not significantly different?
The stiffness values of both sides were averaged for statistical analysis in this study. We used to exam if significant difference existed in sides, and no statistical difference was found.
Line 135: Pictures during the examination would be helpful, especially while standing (supporting frame). Was the supporting frame also during the SWE study?
As suggested, the picture illustrating the assessment in standing position with supporting frame has been added on page 4, “Figure 1. Illustration of the assessment in standing position with a supporting frame.”
Line 137: How was symmetric weight distribution controlled? Just a word command? How can you be sure that the people were standing symmetrically?
Thank you for pointing this out. Subjects were instructed to put their weight equally on both feet and maintain their heads in a neutral position during the measurement. Assessors would also instruct the subjects to adjust posture if subjects standing asymmetrically. Such clarification has been added on page 3-4, line 141-147, stating: “… Subjects were instructed not to resist the compression force generated by TUPS voluntarily, but to reply on the supporting frame instead (Figure 1) … put their weight equally on both feet and maintain their heads in a neutral position during the measurement ... Assessors would instruct the subjects to adjust posture if subjects stood asymmetrically...”
We have also discussed about this and add suggestions on future protocol improvement in Discussion on page 15, line 407-410, stating: “…During the experiment, subjects were instructed to stand symmetrically and put equal weight on both feet, supplemented by the observation of accessors. Future studies can consider to putting pressure sensors and/or pressure mat under subject’s both feet to control the symmetric weight distribution more objectively…”
Results
Line 238: How to explain changes in muscle stiffness in TUPS and their absence in SWE when tested at different levels? The authors explain this by a change in the curvature of the spine. Was there no change in curvature in the SWE study?
We agree that further studies are needed to understand the reason of why no significant difference in SWE was observed. The underlying mechanism of evaluating muscle stiffness is different for TUPS and SWE. While TUPS could not differentiate multiple layers of soft tissues, SWE could specifically locate a certain muscle area by choosing the region of interest (ROI). This could be one possible explanation.
Most previous studies have compared the difference between lying and upright positions and/or between resting and contraction conditions of single muscle group (Young et al, 2020; Koppenhaver et al, 2019). Unfortunately, few studies have used SWE to map the muscle stiffness along the spine. This might be because SWE is a rather new technology and more studies shall be conducted in the future to enable synthesis of information on spinal muscle stiffness as evaluated by SWE.
Such explanation and discussion have been added to Discussion, page 13, line 312-321, stating: “…Unlike TUPS, no statistically significant difference in levels in SWE was observed in this study. This might be explained by the different underlying mechanisms of evaluating muscle stiffness by TUPS and SWE. While TUPS could not differentiate the multiple layers of soft tissues, SWE could specifically locate a certain muscle area by choosing the region of interest (ROI) on ultrasound image. Most previous studies have compared the difference between lying and upright positions and/or between resting and contraction conditions of single muscle group [46, 47]. Unfortunately, few previous studies have used SWE to map the muscle stiffness along the spine, which makes it rather difficult to compare the current findings in this study with previous studies. The lack of previous findings might be because SWE is a rather new technology, and more studies shall be conducted in the future to enable the synthesis of information on spinal muscle stiffness as evaluated by SWE…”
Discussion
Line 265: The authors use the words stiffness and elasticity as synonyms. This is misleading and may be misunderstood.
As suggested, the usage of stiffness and elasticity have been carefully reviewed and revised accordingly throughout the manuscript to avoid the misleading and misunderstood.
Line 266: Is it possible to explain that it is the cause of scoliosis or back pain only because there is greater muscle stiffness at the L4 level?
We agree that the greater muscle stiffness at the L4 level might be one potential cause of scoliosis or low back pain only. Further studies are needed to explore and clarify this. Such statements have been added to page 13, line 293-297, stating: “…The observed large stiffness values at L4 level in healthy adults in this study might help explain the cause/development of the low back pain pathology later on in middle-aged or older adults. It is likely that the greater muscle stiffness at the L4 level might be one potential cause of scoliosis or low back pain, however, future studies shall be conducted to explore and clarify this…”
Line 274-277: If the authors explain the change in muscle stiffness with changes in the curvature of the spine, does it mean that it is a physiological phenomenon? Could this be the cause of pathology?
As suggested, more discussion has been added to page 13, line 293-297, stating: “…The observed large stiffness values at L4 level in healthy adults in this study might help explain the cause/development of the low back pain pathology later on in middle-aged or older adults. It is likely that the greater muscle stiffness at the L4 level might be one potential cause of scoliosis or low back pain, however, future studies shall be conducted to explore and clarify this…”
Page 13, line 310-311, stating: “…It is also worthwhile to investigate if the observed change in muscle stiffness is a physiological phenomenon or the cause of pathology in future studies…”
Page 14, line 376-377, stating: “…The findings of this study inspire future efforts to investigate if the observed change in muscle stiffness is a physiological phenomenon or the cause of pathologies of scoliosis and low back pain…”
Line 288: Elasticity was larger or muscle stiffness was larger?
As suggested, the use of elasticity and stiffness has been carefully reviewed and revised. The description has been revised as “SWE stiffness…” on page 13, line 332.
Line 314: Do these studies really explain the cause and mechanism of scoliosis and back pain?
As suggested, the description has been revised on page 14-15, line 374-385, stating: “…The results of this study will not only enlarge our understanding about the back-muscle stiffness, but also provide us with insights about the possible cause and mechanism of scoliosis and low back pain. The findings of this study inspire future efforts to investigate if the observed change in muscle stiffness is a physiological phenomenon or the cause of pathologies of scoliosis and low back pain. The results may also serve as normative data about back muscle stiffness in healthy young adults, which could be useful in lots of healthcare area in the future. Attention shall be paid to the fact that the normative values might be different for different age groups, and only healthy young people aged between 18-30 were evaluated in this study. The obtained values on muscle stiffness could be treated as a reference, but it might not be appropriate to directly generalize such values to patients with scoliosis or lower back pain. This study can be the first step for further studies investigating the distribution of back muscle stiffness along the spine in different populations, age groups, and patient groups…”
Line 315: I have already written about the normative values. I think that these values will be different for different age groups.
As suggested, such clarification has been added on page 14, line 379-383, stating: “…Attention shall be paid to the fact that the normative values might be different for different age groups, and only healthy young people aged between 18-30 were evaluated in this study. The obtained values on muscle stiffness could be treated as a reference, but it might not be appropriate to directly generalize such values to patients with scoliosis or lower back pain. …”
Line 324: There are no other limitations of this research?
Upon combining the comments from both reviewers, more limitations of this research have been added on page 15, line 407-424, stating: “…During the experiment, subjects were instructed to stand symmetrically and put equal weight on both feet, supplemented by the observation of accessors. Future studies can consider to putting pressure sensors and/or pressure mat under subject’s both feet to control the symmetric weight distribution more objectively. While subjects were instructed to breath naturally during the experiment, it shall be noted that the influence of breath dynamics on the measured results remained unclear. Future studies are needed to understand this issue. The muscle activity was unfortunately not measured as a covariable to explain muscle stiffness changes, which can also be considered in future investigations.
It shall be noted that the SWE and TUPS measurements might be sensitive to other factors beside the mechanical properties. A recent paper reported that indirect tissue stiffness measurements are often sensitive to the mechanical properties, geometrical dimensions, and tensional state of the tissue [54]. From a mechanical perspective, tissue elasticity is defined as ratio of strain (strain = F/A) and elongation (dl/l) and can be measured directly using mechanical measurements. Meanwhile, ultrasound and elastography do not measure strain and elongation directly. These indirect measurements are indirect estimations of the stiffness. When standing, the muscles might contract and thereby influence the stiffness measurements. For the research community, it is essential to have a standard definition and understanding of muscle stiffness. Further studies are still needed to investigate, understand, and clarify this issue.…”
Conclusions
Line 332: The last sentence in the conclusions is an abuse. It does not follow from the results achieved.
As pointed out, the description has been revised on page 15-16, line 429-432, stating: “This may provide some insights regarding the underlying mechanism …and facilitates future research on other spinal musculoskeletal disorders.”

Reviewer 2 Report
Thank you for the oportunity to review this manuscript on mapping of muscle stiffness. I think the manuscript contributes valuable information about stiffness quantification with ultrasound imaging techniques. The manuscript is well written. However, I do have some concerns.
My first concern regards the used term stiffness. In line 49 it is stated that “muscle stiffness can be quantified by tissue elasticity (E)”. From a mechanical perspective, tissue elasticity is defined as ratio of strain (strain = F/A) and elongation (dl/l). Indeed, elasticity can be measured directly using mechanical measurements. Ultrasound and elastography, however, do not measure strain and elongation directly. These indirect measurements are indirect estimations of the stiffness. In a recent paper by Sichting and Kram (Physiol. Meas. 41, 2020), it was shown that indirect tissue stiffness measurements are often sensitive to the mechanical properties, geometrical dimensions, and tensional state of the tissue. With regard to the presented results, it seems possible that SWE and TUPS measurements are also sensitive to other factors besides the mechanical properties. When standing, the muscles might contract and thereby influence the stiffness measurements. I would encourage the authors to include these factors in their discussion and discuss the limitations of indirect stiffness measurements. For the research community, it is essential to have a standard definition and understanding of muscle stiffness.
My second concern regards the two measurement techniques. As described in the manuscript, SWE and TUPS use different methods to quantify muscle stiffness. However, I do miss a discussion on how the different methods contribute to the presented results. For example, TUPS elasticity values range between 104 and 303 kPa, while SWE elasticity ranges between 17 and 37 kPa. How to explain these differences in magnitude? It would be interesting to see if both measurements (SWE vs. TUPS) correlate. There should be a strong correlation if both techniques measure muscle stiffness. It would also be interesting to discuss the possible influences of muscle contraction and overlying tissue on the SWE and TUPS stiffness measurements. Further, the discussion could include a section about best practice recommendations for both techniques.
Besides the two main concerns, I do have some minor comments.
- Concerning the experimental procedure (l.127ff), I wonder if breathing dynamics were considered during the measurements. Breathing might influence the measurements.
- How did the subjects resist the compression force (TUPS) during standing? Did they actively lean against the probe?
- For the results, I would recommend presenting the ICC with 95% confident interval (l.163) (e.g., see Koo and Li, J. Chiropr. Med., 2016).
- The presented results in Figure 1 are confusing. The data for males and females seem not consistent. For example, in the “Standing: female vs. male” plot, the value at L4 close to 300 kPa represents female data (black dot: Female – standing). However, the same value appears in the “Male: lying vs. standing” plot (black rectangle: Male – standing).
- In the discussion (l. 265), I do not fully understand why large stiffness values at L4 in healthy subjects help explaining chronic back pain in this area. Could you please comment on that?
- For the effect of gender (l.281), it would be helpful to explain the difference between male and female subjects.
- Regarding the unexpected finding in the TUPS measurements between males and females (l.304ff), possible reasons could be discussed in more detail. Different anatomical structures as an explanation appears very vague.
- The Limitations section could be expanded. For example, this study did not measure muscle activity as a covariable to explain muscle stiffness changes. Breathing activity was also not considered (at least it is not mentioned in the Methods section).
Author Response
Thank you very much for the efforts and expertise invaluable comments to improve this article. This article has been revised accordingly. Below please kindly find the point-to-point response and detailed description of revisions in the manuscript.
My first concern regards the used term stiffness. In line 49 it is stated that “muscle stiffness can be quantified by tissue elasticity (E)”. From a mechanical perspective, tissue elasticity is defined as ratio of strain (strain = F/A) and elongation (dl/l). Indeed, elasticity can be measured directly using mechanical measurements. Ultrasound and elastography, however, do not measure strain and elongation directly. These indirect measurements are indirect estimations of the stiffness. In a recent paper by Sichting and Kram (Physiol. Meas. 41, 2020), it was shown that indirect tissue stiffness measurements are often sensitive to the mechanical properties, geometrical dimensions, and tensional state of the tissue. With regard to the presented results, it seems possible that SWE and TUPS measurements are also sensitive to other factors besides the mechanical properties. When standing, the muscles might contract and thereby influence the stiffness measurements. I would encourage the authors to include these factors in their discussion and discuss the limitations of indirect stiffness measurements. For the research community, it is essential to have a standard definition and understanding of muscle stiffness.
As suggested, more discussion and clarification have been added to avoid the potential misunderstanding on page 15, line 415-424, stating: “…It shall be noted that the SWE and TUPS measurements might be sensitive to other factors beside the mechanical properties. A recent paper reported that indirect tissue stiffness measurements are often sensitive to the mechanical properties, geometrical dimensions, and tensional state of the tissue [54]. From a mechanical perspective, tissue elasticity is defined as ratio of strain (strain = F/A) and elongation (dl/l) and can be measured directly using mechanical measurements. Meanwhile, ultrasound and elastography do not measure strain and elongation directly. These indirect measurements are indirect estimations of the stiffness. When standing, the muscles might contract and thereby influence the stiffness measurements. For the research community, it is essential to have a standard definition and understanding of muscle stiffness. Further studies are still needed to investigate, understand, and clarify this issue…”
My second concern regards the two measurement techniques. As described in the manuscript, SWE and TUPS use different methods to quantify muscle stiffness. However, I do miss a discussion on how the different methods contribute to the presented results. For example, TUPS elasticity values range between 104 and 303 kPa, while SWE elasticity ranges between 17 and 37 kPa. How to explain these differences in magnitude? It would be interesting to see if both measurements (SWE vs. TUPS) correlate. There should be a strong correlation if both techniques measure muscle stiffness. It would also be interesting to discuss the possible influences of muscle contraction and overlying tissue on the SWE and TUPS stiffness measurements. Further, the discussion could include a section about best practice recommendations for both techniques.
As suggested, to have more information regarding the two measurement techniques, we have conducted the statistical analysis to examine the relationship between the two measurement techniques. The related description has been added to
Method, page 4, line 165-166, stating: “…Pearson correlation test was performed to examine the relationship between the two measurement techniques of SWE and TUPS...”
Results, page 12, line 267-276, stating: “3.5. Relationship between the SWE and TUPS measurement techniques
Table 3 summarizes the results on Pearson’s correlation coefficient (r) between the SWE and TUPS measurement techniques in lying and standing positions. Significantly moderate correlations between the SWE and TUPS measurement techniques were observed at T3 in lying (r=-0.294, p=0.018), L1 in standing (r=0.390, p=0.001), and L4 in standing (r=0.358, p=0.004) positions. Low correlations between the two measurement techniques were observed at the remaining levels and positions, but the correlations were not significant.”
A “Table 3. Relationship between the SWE and TUPS measurement techniques (n=64)” has also been added on page 12.
Discussion, page 14, line 360-372, stating: “4.5. Relationship between the SWE and TUPS measurement techniques
This study observed significantly moderate correlations between the SWE and TUPS measurement techniques at T3 level in lying position, L1 level in standing position, and L4 level in standing position only. This might be caused by the different underlying mechanisms to measure muscle stiffness by the two techniques. Additionally, while TUPS could not differentiate the multiple layers of soft tissues, SWE could specifically locate a certain muscle area by choosing the region of interest (ROI) on ultrasound image. The muscle contraction and the overlying tissue might influence the SWE and TUPS stiffness measurements. The observed correlation may also contribute to the presented results and explain the difference in muscle stiffness as measured by the two techniques at the same location. For example, the TUPS stiffness values ranged between 104 and 303 kPa, while SWE stiffness values ranged between 17 and 37 kPa. This implies that when applying these two techniques to evaluate the muscle stiffness, the relative changes in measured values might be more meaningful for muscle assessment than that of absolute values.”
page 15, line 389-402, stating: “4.7. Best practice recommendations for the SWE and TUPS measurement techniques
Good knowledge in the mechanism of the SWE and TUPS measurement techniques and careful consideration on the available experimental/clinical settings are needed to enable the best practice for the two techniques. It is recommended to apply the SWE to measure the muscle stiffness when differentiating and evaluating different layers of muscles and different regions of the same muscle are needed, since the region of interest (ROI) of SWE can be easily customized to various shapes and sizes at various locations during the measurement. With the advantages of being portable, with wireless connection, and occupying small space, the TUPS would be more helpful to generally evaluate the muscle stiffness at certain anatomical locations, especially for screening purpose in labs/clinics with limited space [33]. The TUPS can be used to do the screenings first, and when abnormal muscle stiffness is identified, the SWE can then be applied to examine the exact cause by specifically locating various regions.”
Besides the two main concerns, I do have some minor comments.
- Concerning the experimental procedure (l.127ff), I wonder if breathing dynamics were considered during the measurements. Breathing might influence the measurements.
As pointed out, more clarification has been added on page 4, line 143-144, stating: “…Subjects were instructed to breath naturally during the experiment…”
Page 15, line 410-412, stating: “…While subjects were instructed to breath naturally during the experiment, it shall be noted that the influence of breath dynamics on the measured results remained unclear. Future studies are needed to understand this issue…”
- How did the subjects resist the compression force (TUPS) during standing? Did they actively lean against the probe?
The subjects were instructed not to resist the compression force generated by TUPS voluntarily, but to reply on the supporting frame instead. Such clarification has been added to page 3-4, line 141-143, stating: “…Subjects were instructed not to resist the compression force generated by TUPS voluntarily, but to reply on the supporting frame instead (Figure 1) …”
A Figure 1 illustrating assessment in standing position with a supporting frame has also been added on page 4.
- For the results, I would recommend presenting the ICC with 95% confident interval (l.163) (e.g., see Koo and Li, J. Chiropr. Med., 2016).
As suggested, the ICC with 95% confident interval is analyzed and presented in Method, page 4, line 161, stating: “…were examined with the Intraclass Correlation Coefficient (ICC (3,1)) with 95% confidence interval (95% CI) …”
Results, page 5, line 175-180, stating: “…Good intra-rater reliability [TUPS: ICC=0.822 (95% CI 0.632 to 0.962) at T7, and ICC=0.905 (95% CI 0.704 to 0.969) at L1; SWE: ICC=0.881 (95% CI 0.631 to 0.962) at T7, and ICC=0.879 (95% CI 0.625 to 0.961) at L1] and good inter-rater reliability [TUPS: ICC=0.742 (95% CI 0.368 to 0.910) at T7, and ICC=0.836 (95% CI 0.602 to 0.943) at L1; SWE: ICC=0.731 (95% CI 0.340 to 0.906) at T7, and ICC=0.781 (95% CI 0.462 to 0.924) at L1] for both measurement instruments were identified in this study (p<0.05)...”
- The presented results in Figure 1 are confusing. The data for males and females seem not consistent. For example, in the “Standing: female vs. male” plot, the value at L4 close to 300 kPa represents female data (black dot: Female – standing). However, the same value appears in the “Male: lying vs. standing” plot (black rectangle: Male – standing).
As pointed out, the presented results in Figure 1 (currently Figure 2) have been carefully reviewed and revised on page 5.
- In the discussion (l. 265), I do not fully understand why large stiffness values at L4 in healthy subjects help explaining chronic back pain in this area. Could you please comment on that?
The chronic back pain commonly affects at L4 level and normally affect the middle-aged or older adults. The observed large stiffness values at L4 level in healthy subjects might help explain the cause of this pathology. To clarify the explanation, more description has been added on page 13, line 293-297, stating: “…The observed large stiffness values at L4 level in healthy adults in this study might help explain the cause/development of the low back pain pathology later on in middle-aged or older adults. It is likely that the greater muscle stiffness at the L4 level might be one potential cause of scoliosis or low back pain, however, future studies shall be conducted to explore and clarify this…”
Page 14, line 376-377, stating: “…The findings of this study inspire future efforts to investigate if the observed change in muscle stiffness is a physiological phenomenon or the cause of pathologies of scoliosis and low back pain…”
- For the effect of gender (l.281), it would be helpful to explain the difference between male and female subjects.
As suggested, more explanations concerning the difference between male and female subjects have been added on page 13, line 326-328, stating: “…Male and female adults have different anatomical structures generally, such as the distribution and mass of body fat and muscles, at the back of trunk [50]. This might help explain the observed difference in back muscle stiffness between the two genders in this study…”
- Regarding the unexpected finding in the TUPS measurements between males and females (l.304ff), possible reasons could be discussed in more detail. Different anatomical structures as an explanation appears very vague.
As suggested, more discussion has been added to page 14, line 349-356, stating: “…The different changes in TUPS stiffness from lying to standing postures between male and female subjects might be due to the different anatomical structures between the two genders. The TUPS used the tissue deformation during the compression-release cycles to acquire the stiffness [33]. It is reasonable to expect that the mass of body fat and breast may affect more of the back muscle elongation and contraction, especially in upper trunk, in female subjects than that of male subjects. Previous studies have also reported increased anterior–posterior shear forces in females and decreased forces in males in response to stress [52], and different trunk muscle geometry between the two genders [53]. All these findings might help explain the measured different changes from lying to standing posture between the SWE and TUPS measurements and between the two genders…”
- The Limitations section could be expanded. For example, this study did not measure muscle activity as a covariable to explain muscle stiffness changes. Breathing activity was also not considered (at least it is not mentioned in the Methods section).
As suggested, the Limitation section has been expanded on page 15, line 404-424 stating: “…While mapping the muscle stiffness at thoracic and lumbar regions could already provide plenty of information, the data on cervical and sacral regions were not involved in this study. Future studies could consider to expanding this mapping strategy to provide a more comprehensive picture of spinal muscle stiffness. During the experiment, subjects were instructed to stand symmetrically and put equal weight on both feet, supplemented by the observation of accessors. Future studies can consider to putting pressure sensors and/or pressure mat under subject’s both feet to control the symmetric weight distribution more objectively. While subjects were instructed to breath naturally during the experiment, it shall be noted that the influence of breath dynamics on the measured results remained unclear. Future studies are needed to understand this issue. The muscle activity was unfortunately not measured as a covariable to explain muscle stiffness changes, which can also be considered in future investigations.
It shall be noted that the SWE and TUPS measurements might be sensitive to other factors beside the mechanical properties. A recent paper reported that indirect tissue stiffness measurements are often sensitive to the mechanical properties, geometrical dimensions, and tensional state of the tissue [54]. From a mechanical perspective, tissue elasticity is defined as ratio of strain (strain = F/A) and elongation (dl/l) and can be measured directly using mechanical measurements. Meanwhile, ultrasound and elastography do not measure strain and elongation directly. These indirect measurements are indirect estimations of the stiffness. When standing, the muscles might contract and thereby influence the stiffness measurements. For the research community, it is essential to have a standard definition and understanding of muscle stiffness. Further studies are still needed to investigate, understand, and clarify this issue...”

Round 2
Reviewer 1 Report
Thank you for the next opportunity to review the article entitled “Mapping of back muscle stiffness along spine during standing and lying in young adults: A pilot study on spinal stiffness quantification with ultrasound imaging”. In my opinion, the authors have corrected the manuscript in accordance with the recommendations. So I have no more comments and I think that the work should be accepted for publication.